# Synthesizing lesions using contextual GANs improves breast cancer classification on mammograms

**Author(s) names withheld**                                    EMAIL(S) WITHHELD

## Abstract

Data scarcity and class imbalance are two fundamental challenges in many machine learning applications to healthcare. Breast cancer classification in mammography exemplifies these challenges, with a malignancy rate of around 0.5% in a screening population, which is compounded by the relatively small size of lesions ($\sim$1% of the image) in malignant cases. Simultaneously, the prevalence of screening mammography creates a potential abundance of non-cancer exams to use for training. Altogether, these characteristics lead to overfitting on cancer cases, while under-utilizing non-cancer data. Here, we present a novel generative adversarial network (GAN) model for data augmentation that can realistically synthesize and remove lesions on mammograms. With self-attention and semi-supervised learning components, the U-net-based architecture can generate high resolution (256x256px) outputs, as necessary for mammography. When augmenting the original training set with the GAN-generated samples, we find a significant improvement in malignancy classification performance on a test set of real mammogram patches. Overall, the empirical results of our algorithm and the relevance to other medical imaging paradigms point to potentially fruitful further applications.

## 1. Introduction

Common to many machine learning applications in healthcare, developing algorithms for breast cancer detection in mammography (11; 14; 15; 8; 18; 9; 10; 12) is heavily prone to overfitting given the difficulty in collecting large amounts of positive examples. A malignancy prevalence of around 0.5% in a screening population leads to a stark class imbalance, which is exacerbated by the fact that malignant lesions can be subtle and typically only occupy a small area relative to the surrounding normal-appearing breast tissue. On the other hand, non-malignant mammograms can be relatively abundant, but tend to be underutilized in machine learning approaches, as overfitting can occur rapidly on the cancer examples during training. Given the success of standard data augmentation strategies in combating overfitting, recently there have been numerous efforts exploring the use of generative adversarial networks (GANs) (19) for data augmentation (24; 25; 26; 27; 28). While baseline GAN-augmented training methods may not be effective for natural images (29), mammography specifically lends itself well to structured approaches (13). Here, we present a novel GAN model designed to synthesize and remove lesions on mammograms. Importantly, instead of creating new training examples from scratch, the approach relies on the biological intuition that lesions can develop approximately uniformly across breast tissue. Given the context of surrounding tissue, the proposed model is able to realistically generate and remove lesions. We demonstrate that,

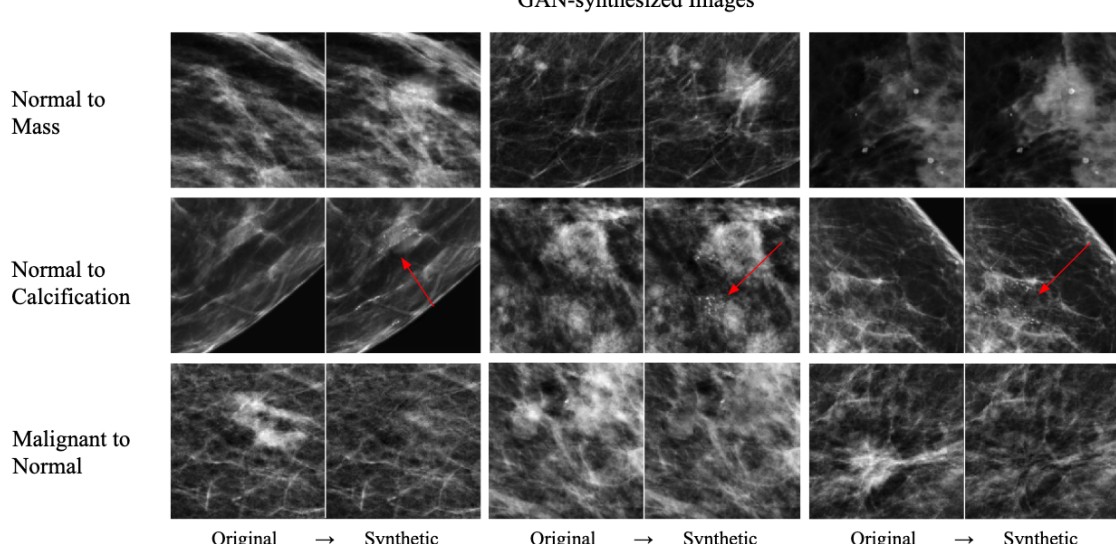

Figure 1: Examples of synthesized and removed lesions by the proposed GAN architecture. Each pair of images represents a real image (left) and the synthetic counterpart (right). Each row represents a different synthesis task. Zooming is encouraged for viewing the synthesized calcifications (row 2).

as a data augmentation procedure, the approach leads to a meaningful boost in classification performance for the presence of breast cancer in mammogram image patches.

The paper is structured as follows. We begin by describing the architecture of our proposed contextual GAN. Using a series of optimized components, we demonstrate that the model is capable of generating high resolution (256x256px) mammogram patches, where lesions are either generated on or removed from surrounding tissue. Next, we illustrate the effect of including the GAN-outputted patches in a training set of mammogram patches, where a ResNet-50 (22) classifier is used to classify the presence or absence of cancer in the patches. Testing on a set of held-out real patches, we find that the GAN augmentation leads to improved performance for a range of real/generated sampling ratios. Finally, we visualize the feature embeddings of the real and generated data to gain further insight into the realized results.

## 2. Proposed Contextual GAN Architecture

Figure 2 describes the architecture of the proposed generator network. The model uses a U-net (17) design, which encodes the input image and generates (or removes) lesions using skip connections from the intermediate encoding layers. The generator takes as input an image with size 256x256px, and consists of an encoding (blue bars in Figure 2) network and a decoding network (in yellow). The encoding component starts with 16 filters at the first convolutional layer block, and doubles the number of filters and halves the spatial resolution per block. Each block consists of a concatenation of the input to the block and a random scalar value (drawn uniformly from [-1,1] and reshaped to a 1x1 pixel and resized to match the

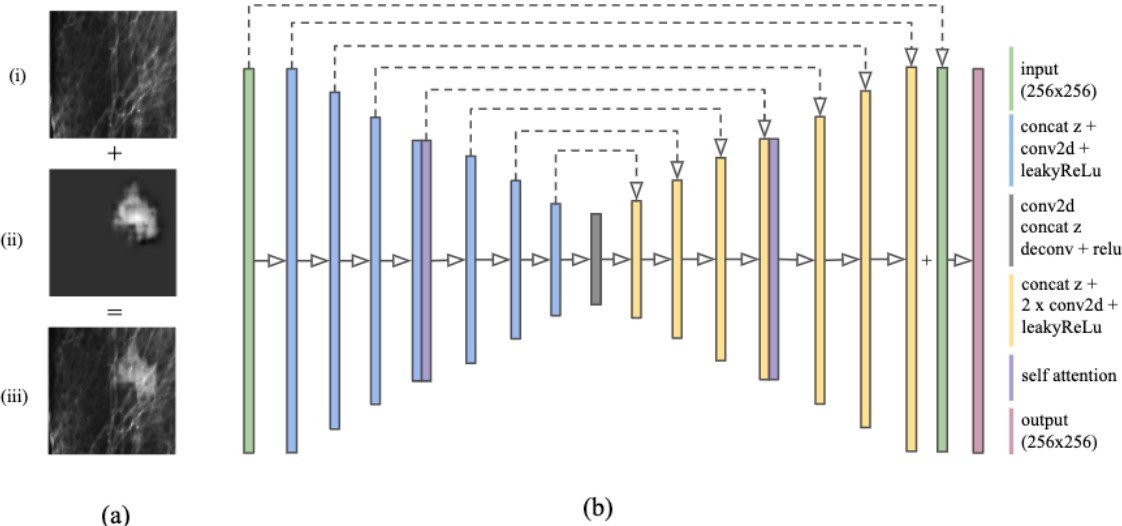

Figure 2: The GAN generator architecture. In (a), the input image (i) is fed into the generator, which produces the output (ii) that is added with (i) to produce the finalized synthetic patch (iii). The diagram (b) describes the generator architecture. The discriminator is essentially the encoding part of the network, but with 8x the number of filters at each layer and max pooling instead of strides of 2.

input dimensions), followed by a convolutional layer with stride size of 2 and 3x3 kernel, and a LeakyReLU (16) activation function. Akin to other GAN approaches, the random scalar value is used as a sampling procedure to allow the generator to produce a variety of options given one input image. At the 32x32 resolution, we use a self-attention module (4), which is also used in the encoding part of the generator and discriminator. At the central layer of the generator, a 4x4x2048 tensor is compressed into a 1x1x4 embedding, which forms the input for the decoding part of the generator. Each decoding block consists of concatenating the skip connection and random scalar value to the input, followed by an up-sampling operation (nearest neighbor) and two convolutional layers (with ReLU activation). The output of the final block at 256x256px resolution is passed through a final 1x1 convolutional layer, followed by a 10px border cropping and clipping of values within [-1.0, 1.0]. The resulting output is then added to the original input image (described in more detail below). The discriminator is identical to the encoding part of the generator, but with 8x the number of filters per layer and strides of 2 replaced with max pooling operations. We train separate models for generating masses, calcifications, and removing lesions. While we experimented with a conditional formulation, where one central model was used along with a category-specific input label, we found that training separate models proved to be the most stable. We describe additional architectural details below, along with the training and loss formulations.

**Separate lesion and image channels.** A unique feature of the proposed generator is that the neural network output only synthesizes the lesion. This lesion is then added to

the input image to form the combined output image. The lesion, input, and combined images are concatenated to form a final three-channel output image. This approach greatly stabilizes the training process by allowing the generator to focus on just synthesizing the lesion. Additionally, we apply several post-processing steps to the lesion channel (as described below in "Synthesizing dataset") to further refine the output.

**Semi-supervised training loss.** To further encourage malignant features to be generated in the samples, we utilize a semi-supervised loss formulation (1). We extend the binary cross entropy loss of [real (malignant), fake (malignant)] to include benign and normal patches as well for a four-way output of [real, fake, benign, normal]. This approach allows the discriminator to penalize examples containing benign or normal features. During training, the discriminator is given examples of all four classes for each update step, with the losses from benign and normal examples scaled by a factor of 0.2. For additional stability, a gradient penalty was added to the discriminator loss as detailed in (6):

$$\lambda \cdot E_{x \sim P_{real}, \delta \sim U(0,1)} \left[ ||\nabla_x D_\theta(x + \delta)|| - k \right]^2$$

where $\lambda = 10$, and $k = 1$.

**Progressive growing.** Generating patches at 256x256px resolution in one shot results in frequent mode collapse, so we used progressive growing (2) from 128px to 256px, which achieved significantly more stable and high quality results. The generator is first trained to produce images at 128px resolution. Then, a new layer is appended to the generator to produce images at 256px. We slowly fade in the new layer by doubling the resolution of the previous 128px layer using nearest neighbor and linearly blending with the 256px layer. Over 3000 iterations, the weight on the 128px layer decreases from 1 to 0 while the weight of the 256px increases from 0 to 1. The discriminator follows an identical but reversed schema.

**Self-attention module.** We used self-attention (4) modules in both the generator and discriminators, as well as spectral normalization (5). Self-attention has been shown (4) to improve recognition of long-range dependencies and utilize features from the entire patch.

**Border cropping.** A 10px border around the generated lesion mask is cropped out to smooth border artifacts. We find that this technique alone is effective in matching the features of the input image, and that common techniques like L1 loss or perceptual loss are unnecessary.

**Training details.** We use the Adam (3) optimizer with a learning rate of $1e^{-5}$, and $\beta_1 = 0.0$, $\beta_2 = 0.99$, $\epsilon = 1e^{-8}$ for both the generator and discriminator. For masses, we train the generator twice for every one iteration of the discriminator for better convergence. Images are clipped to a $[-1, 1]$ pixel value range.

## 3. Experiments

**Dataset details.** For training and evaluation, we use the Optimam Mammography Image Database (21), a publicly obtainable dataset from a large screening population in the UK. Our dataset contains 8,282 images with a malignant lesion, 1,287 images with a benign lesion,

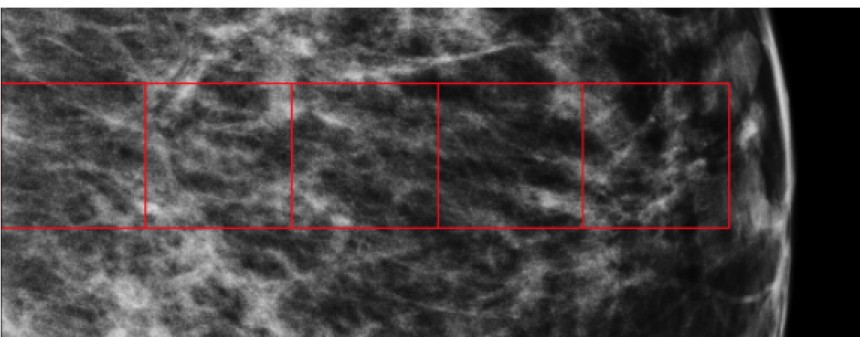

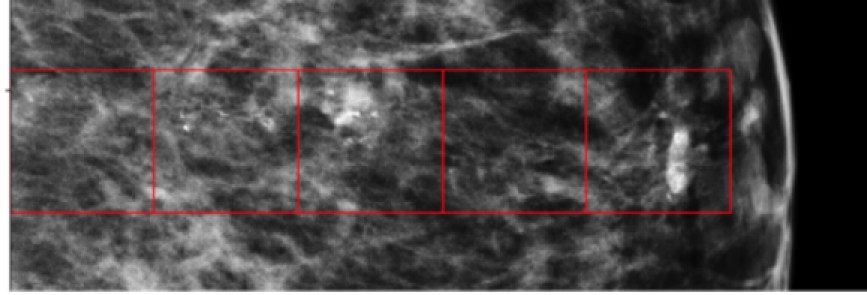

Figure 3: Demonstration of GAN synthesis on contiguous boxes in a mammogram. A section of a normal mammogram with five 256x256px patches in a row is selected for augmentation to illustrate how the GAN works in varying contexts (above). The GAN synthesizes a lesion onto each patch, and the patches are then reinserted back into the mammogram (below).

and 16,887 normal studies. The data is split into 60%/20%/20% training/validation/testing splits. Approximately half of the cancer and benign cases contain radiologist-annotated bounding box labels. From these full images, we generated patches according to the following quantities: for training, we created 400,000 normals, 42,280 malignant masses, 3,500 benign masses, 24,100 malignant calcifications, and 6,580 benign calcifications from full images in the training split; for validation and testing (each), we created 1,000 normals, 1,000 malignant masses, and 1,000 malignant calcifications from full images in the validation and testing splits. To create patches, a random location is picked on a random mammogram image that contains at least 90% breast tissue. Then, the patch is randomly flipped (left/right), rotated (90, 180, or 270 degrees), and resized (uniformly from 0.8 to 1.2x) for augmentation. To specifically create patches containing malignant lesions, a random pixel on the lesion is chosen, and then a random x and y offset is chosen between 0 to 128 pixels in either direction.

**Synthetic dataset.** To create a synthetic lesion example, we follow the following procedure:

1. Randomly sample a healthy mammogram image.

2. Randomly sample a 256x256px patch that contains more than 90% breast tissue.

| Training Regime | AUC | P-value |
|:---:|:---:|:---:|
| Baseline | 0.829 | - |
| 10% w/ decay | **0.837** | 0.10 |
| 25% w/ decay | **0.839** | 0.055 |
| 50% w/ decay | **0.846** | 1e-4 |
| 75% w/ decay | 0.828 | 0.72 |
| 100% w/decay | 0.797 | 1e-8 |

Table 1: Experimental results comparing model performance with and without GAN-augmented data. A baseline model trained on only real images is run alongside models given varying starting rates of GAN-augmented training examples. The highest performing GAN-augmented model yielded an improvement of 0.017 AUC over the baseline.

3. Input the patch into the generator and generate the synthetic patch.

4. Perform the following steps to post-process the synthetic patch:

   - Isolate the lesion channel in the synthetic patch.
   - Create a binary mask of the channel by applying a threshold of 0.1.
   - Use connected-component labeling and pick the largest object in the binary mask.
   - Discard the object if it is too small ($< 10\%$ of the patch area).
   - Expand the edges of the object by 5px.
   - Apply a 10px Gaussian smoothing filter to the edge of the object.
   - Element-wise multiply the resulting mask with the lesion channel.

5. Add the lesion channel to the base normal patch.

We found that the post-processing steps removes background noise from the image while still retaining the relevant synthetic lesion. For removing lesions from malignant mammograms, we apply the same steps but use a $< -0.1$ binary pixel threshold. The extracted lesion is then treated as a 'negative lesion', which when added back into the input image removes the lesion. To create our synthetic training dataset, we generated 5000 examples each of mass, calcification, and normal patches.

**Patch Classifier Training.** For a patch classification model, we use ResNet-50 (22), initialized with weights trained from ImageNet. We train the ResNet-50 for a binary cancer/no-cancer task using different proportions of synthetic data compared to real data. In each case, positive and negative examples are sampled with equal probability. Models are trained for 500K samples, using the Adam optimizer (23) with a learning rate 1e-5, $\beta_1 = 0.9$, and $\beta_2 = 0.999$. When including synthetic data, an initial proportion is chosen, and then decayed by 10% every 5000 training samples. A decay was used to mimic a "fine-tuning"

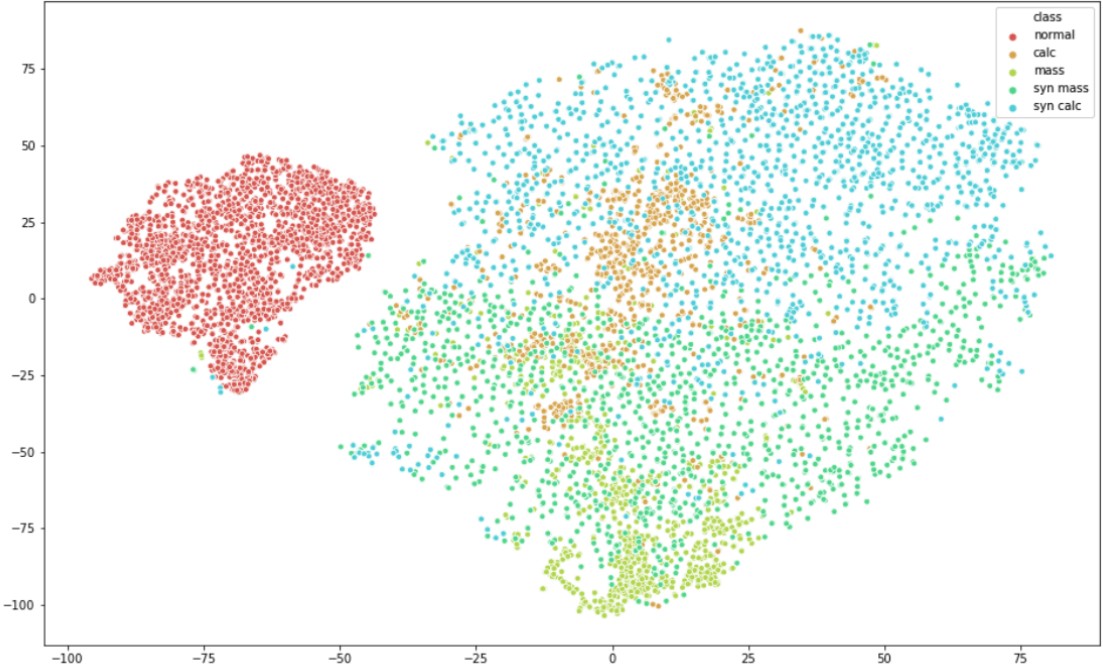

Figure 4: A plot of the t-SNE embedding using the last feature layer of the ResNet-50 classifier, trained only on real mammogram patches. The red points represent real normal examples, the orange points represent real malignant calcifications, the green points represent real malignant masses, the blue-green points (bottom) represent synthetic malignant masses, and the blue points represent synthetic malignant calcifications (top). As illustrated, the embeddings for the synthetic lesions cluster with real lesions, even using the embeddings from a classifier trained on only real lesions.

scenario, where the model was trained on larger proportions of real data over time. For initial proportions of synthetic data, we use 0% (all real), 25%, 50%, 75%, and 100%. In each case, the final weights are chosen based on performance on the validation set (all real), as evaluated every 5000 samples.

**Results.** The results of our experiments are shown in 1. As the percentage of synthetic data initially is increased, test performance on real data rises to a peak of 0.846 AUC at a 50% initial synthetic rate (compared to 0.829 AUC trained only on real data; $p < 1e - 8$). The improvement in performance was significant for rates of 10%, 25%, and 50%. Beyond a 50% initial synthetic rate, the test performance on real data declines. P-values are computed using the DeLong method (20).

**t-SNE Embedding.** To better understand how the real and synthetic data are clustered and the effect of the augmented model, we performed a t-SNE embedding analysis. From the validation data, we sampled 2000 normal, 1000 real malignant mass, 1000 real malignant calcifications, 3560 synthetic malignant mass, and 4000 synthetic malignant calcifications

patches. We inputted each patch through the baseline ResNet-50 model trained only on real patches to obtain a feature vector, using the last feature layer in the model. A two-dimensional t-SNE embedding plot using these features is shown in Fig. 4. A first feature to note is that the normal patches appear clustered away from the real mass and real calcifications patches. This is to be expected, given that the ResNet-50 model was trained on real data. Interestingly, the synthetic mass and calcifications patches are also clustered away from the normal patches and generally overlap with the real lesion patches in the embedding. While it may be expected that this would be the case, it is not guaranteed and provides further support that the GAN is generating features that are consistent with real lesions, as dictated by the features learned to distinguish between normal and malignant patches. In the appendix, we also illustrate how several real false negatives that are originally clustered towards the real patches in the embedding become correctly classified and subsequently cluster towards the real lesions after GAN-augmented training (Fig. 5).

## 4. Discussion

Deep learning for cancer classification in mammography has shown promising progress, yet data scarcity and class imbalance is still a significant roadblock to its continued progress. In this paper we explore the use of GANs as a data augmentation technique for classification networks. Using a U-net based architecture with self-attention and semi-supervised learning, we synthesize lesions onto normal-appearing mammogram patches and remove lesions from patches where they are present. We demonstrate that incorporating synthetically augmented mammogram patches into the training regime improves overall model performance. Without additional data or any changes to the underlying architecture, the GAN-augmented regime produced an AUC of 0.846, 0.017 greater than the baseline. Through visualizing a t-SNE embedding of the last classifier layer, we observe that the synthetic data distribution covers and expands upon the original training data distribution. As a future step, we aim to extend our GAN formulation toward improving full image object detection networks by reinserting synthetic patches back into the full image and providing the bounding box for localization. Code will be made available on Github before publication. Overall, our contextual GAN model and data augmentation results show promise for rectifying data imbalance in mammography, and can be adapted to address similar issues in other medical imaging domains.

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

## 5. Appendix

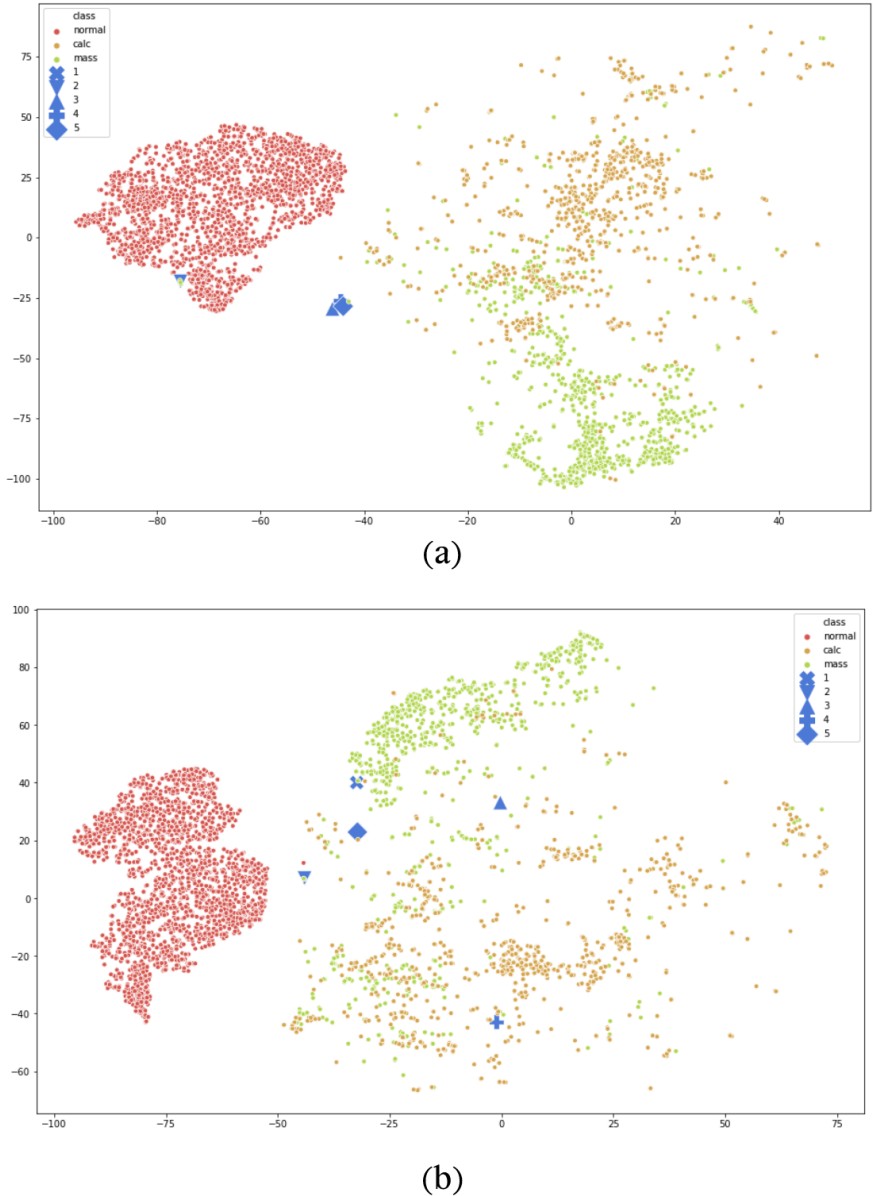

(a)

(b)

Figure 5: Using the t-SNE embedding from Figure 4, we highlight five misclassified malignant examples (in dark blue). (a) displays the proximity of these points to the normal cluster from the embedding produced by the baseline model. (b) shows the same points moving toward the malignant clusters and away from the normal cluster when using the embedding produced by the augmented model and increase in malignant classification score by an average of 0.32.

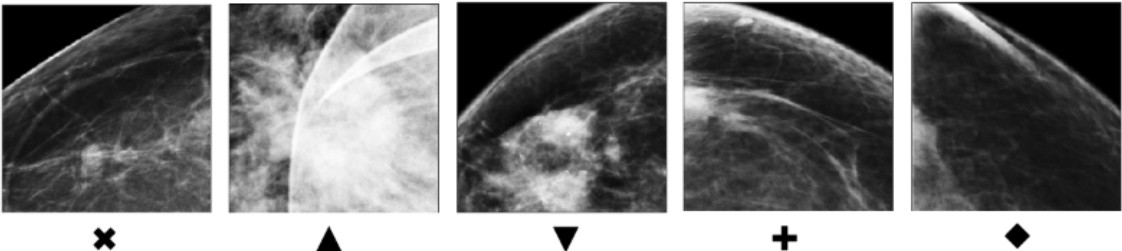

Figure 6: Image patches containing malignant lesions referenced by the blue data points in Figure 5. These patches scored low ('normal') with the baseline model, but scored high ('malignant') with the augmented model.

