# OpenReview forum: "Synthesizing lesions using contextual GANs improves breast cancer classification on mammograms"
_MIDL.io/2020/Conference — Submitted to MIDL 2020_

### Official Review · AnonReviewer1 · 2020-02-24
**A GAN based approach for data augmentation albeit with complicated architecture without ablation experiments warranting choice of the methodology**

**Rating:** 2
**Confidence:** 5

**Summary:**

The novel contribution of the paper is a GAN based approach for data augmentation in mammography images to aid in the classification of benign vs. malignant breast lesions. Given the pervasive problem of lack of datasets especially with balanced proportion of lesions vs. non-lesions in medical image sets, an approach to produce realistic augmented sets with variations from the original data is highly relevant and important for improving the generalization capability of the models trained in medical image sets.

Pros: 1. While GAN based augmentation itself is not new (e.g. Choi et.al ICCV 2019), GAN based data augmentation in this application is new.
2. The presented visual result seems  qualitatively convincing.
3. Experiment to assess the value of augmented sets as proportion of augmented vs. real set is useful.

Cons:
1. There isn't anything methodologically new in this approach. The use of semi-supervised classification (Odena), gradient penalty loss, self-attention modules have all been used although in other contexts.
 2. A bigger issue is that the approach seems rather over-complicated. Particularly, the choice of three different networks; authors say that use of the z variable to select between the different categories didn't work, but why this didn't work is not explained. I wonder if this has to do with how the training was done, or with the use of an over-complicated network there wasn't enough data for obtaining generalizable training. Experiments to explain this choice is necessary.
3. What is the use of attention module is not clear. It seems like its just an add on that would not really have an useful effect. From the shown examples, small patches are selected and a random location is chosen for adding lesions or calcifications. What is attention module doing here? You don't really need it for anything. This should be explained and backed up with ablation experiments.
4. Lesion removal explanation is very confusing. It almost reads like you do a lesion detection and then remove it. Is this correct? If that's the case, why then do you need a GAN? You could just do it with a lighter weight and easier to optimize feedforward segmentor. Please explain and clarify this method.


**Strengths:**

A GAN based data augmentation approach applied to mammography image classification is potentially useful. Given the pervasive problem of lack of datasets especially with balanced proportion of lesions vs. non-lesions in medical image sets, an approach to produce realistic augmented sets with variations from the original data is highly relevant and important for improving the generalization capability of the models trained in medical image sets.



**Weaknesses:**

The paper is lacking in a lot of details related to the method, e.g. the self-attention implementation, spectral normalization - this is just mentioned but never expanded. The implementation and training details could be expanded a bit more.

There isn't anything methodologically new in this approach. The use of semi-supervised classification (Odena), gradient penalty loss, self-attention modules have all been used although in other contexts. The rationale for semi-supervised classification is briefly presented but not validated in the results. The same goes for the gradient penalty loss, and self-attention modules. Why do you even need a self-attention in these kinds of images. There doesn't seem to be any specific region other than the one already indicated randomly to produce a lesion image. Gradient penalty is mostly to prevent mode collapse issues. How stable is the convergence with and without gradient penalty. These results should be presented.

Lesion removal method is confusing or wrong.

Also comparison to other approaches such as GAN-based methods are missing. Why is this method better and what aspects or modules used in this architecture offer improvements.

**Detailed Comments:**

Please see my summary comments and weakness/rebuttal suggestions.

**Justification Of Rating:**

There isn't anything methodologically new. The paper combines several well-known techniques. Although this by itself is not a strong negative, the rationale for adding the individual components to produce such a seemingly over-complicated data augmentation approach should be validated with experiments. Also comparative experiments to other GAN-based data augmentation methods would be helpful. The lack of such experiments makes this paper less exciting.

**Paper Type:**

both

**Questions To Address In The Rebuttal:**

Rationale for the chosen architecture needs to be presented. Its unclear and just reads like a mish-mash of some of the latest methods.

Ablation experiments are hard to perform in the short rebuttal period. But the paper reads like certain design choices were made based on some of the preliminary observations. Ex. the use of multiple networks instead of one for generating the different types of images. Experiments showing these comparisons are useful. Also details of the training and testing should be expanded on to understand why this happened and to clarify that it is not just result of using a network with much larger number of parameters.

Lesion removal method should be clarified. It seems wrong.

**Special Issue:**

no

---

> ### Author Response · Authors · 2020-03-28
> **Thank you for your review**
>
> Dear Reviewer,
>
> We appreciate you taking the time to review our paper, and we hope that our responses sufficiently address your questions.
>
> 1. “Why do you even need a self-attention in these kinds of images. There doesn't seem to be any specific region other than the one already indicated randomly to produce a lesion image.”
>
>     This is an instructive question and one in which we will comment upon in the revision. Within the sampled 256x256px patch, there is still the question about the location and phenotypic features of the lesion. Whereas previous approaches like [13] randomly sample this according to a uniform prior, introducing a self-attention mechanism allows the network to incorporate information from the entire patch rather than just local features. As a result, we find that the lesions produced are more spatially coherent.
>
> 2. “Gradient penalty is mostly to prevent mode collapse issues. How stable is the convergence with and without gradient penalty.”
>
>     That is correct, we used gradient penalty and spectral normalization to prevent mode collapse. In our experiments, using both resulted in more stable training, as visualized here: https://www.dropbox.com/s/pfm67w2ptxow154/loss.png?dl=0. “D_loss” and “g_loss” represent the discriminator and generator loss across 50K iterations, respectively. With gradient penalty and spectral normalization, the generator and discriminator losses increase and decrease at a slower rate versus the baseline, indicating more stable training.
>
> 3. “Lesion removal method is confusing or wrong”
>
>     We apologize for the confusion and will clarify in the revision. In the case of removal, the generator outputs a pixel mask, which works as “white-out” over the lesion. When added to the base image, the lesion is covered up by this pixel mask. The lesion pixel values range from [-1.0, 1.0], so setting a threshold of <-0.1 allows the mask to subtract from the overall intensity of the lesion.
>
> 4. “Also comparison to other approaches such as GAN-based methods are missing. Why is this method better and what aspects or modules used in this architecture offer improvements.” and “Rationale for the chosen architecture needs to be presented. Its unclear and just reads like a mish-mash of some of the latest methods. Ablation experiments are hard to perform in the short rebuttal period. But the paper reads like certain design choices were made based on some of the preliminary observations. Ex. the use of multiple networks instead of one for generating the different types of images. Experiments showing these comparisons are useful.”
>
>     Thank you for the opportunity to provide more clarity into the architectural choices. The primary architecture contribution of the paper is the U-net architecture with channel separation. Whereas previous approaches like [13] only use a random seed to generate the lesion, the U-net architecture with self-attention allows the network to incorporate contextual information to produce more natural lesions. Channel separation is used to stabilize training by allowing the generator to focus on generating only the lesion and not the surrounding context. The other methods used, like gradient penalty, spectral normalization, and progressive growing, are also used to further stabilize training (as shown in the link above: https://www.dropbox.com/s/pfm67w2ptxow154/loss.png?dl=0). Finally, multiple networks are used for the simple reason that a combined network does not reliably produce calcifications, but separate networks do.
>
> 5. “Also details of the training and testing should be expanded on to understand why this happened and to clarify that it is not just result of using a network with much larger number of parameters.”
>
>     To clarify, in the classification task, the same network (ResNet-50) is used for the baseline and GAN-augmented regimes, and the only difference in regimes are the number of synthetic examples used, not the size of the network. Regarding the GAN network size, the primary increase in parameters is in the encoding component of the generator. This allows the network to incorporate contextual information, and would not be achievable in networks with only a decoding component like [13] even with merely an increased number of parameters.

---

> ### Comment · Area_Chair1 · 2020-03-28
> **Rebuttal**
>
> Dear Reviewer
>
> Can you read the rebuttal and see if it clarified the issues identified in your review?
>
> Thanks
> Your AC.

---

> > ### Comment · AnonReviewer1 · 2020-04-09
> > **Convincing results to demonstrate why attention is necessary.**
> >
> > Thank you for the detailed response.
> > "Whereas previous approaches like [13] randomly sample this according to a uniform prior, introducing a self-attention mechanism allows the network to incorporate information from the entire patch rather than just local features. As a result, we find that the lesions produced are more spatially coherent."  -- while the rationale seems plausible, its not convincing because these images really don't seem to depict any spatial organization that would warrant the use of such a method. Perhaps if visual results were presented along with comparison to uniform prior based methods, that might be more convincing. As I did not see those, I am inclined to believe that attention methods are still not useful here.
> >
> > Given this, I am still not convinced and cannot change my initial assessment.

---

### Official Review · AnonReviewer4 · 2020-03-12
**Very Interesting idea but lacks sufficient support**

**Rating:** 2
**Confidence:** 5

**Summary:**

This work used a U-net like structure to GAN to add/remove lesions from healthy/cancerous mammogram. In this work, the authors add self-attention and apply progressive GAN training to achieve a proper result in their standard. It is an innovative way to tackle the scarcity in lesions of mammography by utilizing a large amount of healthy data. The result shows that the performance is at least on par with the baseline they choose.

**Strengths:**

1. Synthesize proper lesions instead of synthesizing mammograms with lesions is a great way to simplify the targeted issue.
2. Extending the binary cross-entropy loss of the discriminator to a four-way output is an interesting way to perform class-wise discrimination.


**Weaknesses:**

1. The threshold 0.1 in the post-processing seems important. However, there is no clear explanation that why they choose this value. It is more persuasive to do a sensitive study on the selection of the threshold.
2. t-SNE embedding does not reveal anything in the term of how data points distributed in hyperspace Showing this embedding would not help to disclose the effect of the augmented model. It is only an intuitive tool to show how data would cluster in a high dimension.
3. The authors do not discuss why would the performance peak at 50% w/decay or the effect of synthetic image rate that included in the training data.


**Detailed Comments:**

1. In this work, the authors synthesize calcification, mass, and normal patches. However, in the experiment, they use a binary label. So I wonder how they include images in the dataset. Is it normal - no-cancer, mass and calcification - cancer or other cases? Please clarify.
2. In the paragraph “Progressive growing”, the last line. “The discriminator follows an identical but reversed schema.” Is it a typo that ‘schema’ should be ‘scheme’?
3. On page 6, the last line “decayed by 10% every 5000 training samples”. Is it a typo that ‘samples’ should be ‘steps’? or other meanings? Confused.
4. Please show a synthetic lesion before and after post-processing if possible. It would be very interesting to know how well ‘raw’ lesion the network can synthesize.
5. Please clarify what happens to decay? If 10% of synthetic data is decayed, does it mean it is replaced by real data?
6. The P-value shown in this table is unclear. Please clarify what it stands for and what kind of test is performed here.


**Justification Of Rating:**

I find this work is promising but limited in the validation and some analysis in the result. I hope the authors can improve what left behind. If this work is fairly evaluated, it would contribute to the topic of the image synthesizing in the medical field.

**Paper Type:**

validation/application paper

**Questions To Address In The Rebuttal:**

1.If possible, please do a sensitivity study on point 1 in the weakness show the effect of the threshold.
2.Please discuss a bit about point 3 in the weakness.


**Special Issue:**

no

---

> ### Author Response · Authors · 2020-03-28
> **Thank you for your review**
>
> Dear Reviewer,
>
> We appreciate you taking the time to review our paper, and we hope that our responses adequately address your questions.
>
> 1. “Please do a sensitivity study on point 1 in the weakness show the effect of the threshold.”
>
>     We agree that presenting more detailed analysis of the threshold would be useful. In our experiments, we found that the threshold was primarily useful for removing low intensity background noise, namely values much less than 0.1. In fact, for relatively small values around 0.1, the effect of threshold is hardly visible. We have included a visualization (https://www.dropbox.com/s/gt4n1k5qxv4eqcw/threshold.png?dl=0) that we will add as a supplemental figure to demonstrate the effect of the threshold value on the image. The first column represents the base image that is fed into the generator. The second column is the lesion channel only. The 3rd to 9th columns are the post-processed images at various thresholds (a black image indicates that the threshold removed too much lesion that the example was discarded). In general, thresholds less than 0.1 showed minor differences, and thresholds above 0.1 often removed too much from the lesion mask.
>
> 2. “The authors do not discuss why would the performance peak at 50% w/decay or the effect of synthetic image rate that included in the training data.”
>
>     While it may be difficult to predict a priori that performance would peak at 50% w/ decay, one might intuit that performance would peak at moderate ranges of synthetic to real ratios. On one hand, if the percentage of synthetic images included is very low, it might not have much of an effect beyond the baseline scenario. At the other extreme, very high synthetic ratios may cause the classifier to overfit on the synthetic images and not learn a generalizable representation to real images. Thus, one might imagine that a moderate mix would lead to the best performance. We empirically find that a 50% ratio is the optimal point. In the revision, we will add these points to the discussion.
>
> 3. “In this work, the authors synthesize calcification, mass, and normal patches. However, in the experiment, they use a binary label. So I wonder how they include images in the dataset. Is it normal - no-cancer, mass and calcification - cancer or other cases? Please clarify.”
>
>     Yes, that is correct. To clarify, both benign and normal cases are classified as “no cancer”, while malignant masses and calcifications are classified as “cancer”.
>
> 4. "In the paragraph “Progressive growing”, the last line. “The discriminator follows an identical but reversed schema.” Is it a typo that ‘schema’ should be ‘scheme’?"
>
>     Thank you, we have corrected this to ‘scheme’.
>
> 5. "On page 6, the last line “decayed by 10% every 5000 training samples”. Is it a typo that ‘samples’ should be ‘steps’? or other meanings? Confused."
>
>     To clarify, each step consists of training on one sample (batch size of one), so “steps” and “samples” are essentially equivalent here. We have clarified this in the revision.
>
> 6. "Please show a synthetic lesion before and after post-processing if possible. It would be very interesting to know how well ‘raw’ lesion the network can synthesize."
>
> Thank you for this suggestion. In the link referenced above (https://www.dropbox.com/s/gt4n1k5qxv4eqcw/threshold.png?dl=0), we have included examples of before and after post-processing. The ‘raw’ lesion is actually quite visually similar to the post-processed lesion in most cases; we find that this post-processing step is actually most useful for throwing out examples where the lesion did not successfully synthesize (i.e., where the lesion is too small, like in the 4th row of the included figure -- this example was filtered out and not used in training).
>
> 7. "Please clarify what happens to decay? If 10% of synthetic data is decayed, does it mean it is replaced by real data?"
>
>     Yes, when we decay, the proportion of real data increases, while the proportion of synthetic data decreases. If 10% of synthetic data is decayed, then we start with synthetic data being 10% of the training samples (and real data being 90%), then decay to 9% after 5K steps, 8% after 10K steps, etc. until 100% of the training data is real and no synthetic data is used in training.
>
> 8. "The P-value shown in this table is unclear. Please clarify what it stands for and what kind of test is performed here."
>
>     We apologize for the lack of clarity and will make sure to clarify this in the caption for the revision. The P-values are computed using the DeLong method, which is a statistical test for comparing two AUCs. The P-values are calculated by comparing the given training regime with the baseline AUC.

---

> ### Comment · Area_Chair1 · 2020-03-28
> **Review after rebuttal**
>
> Dear Reviewer
>
> Can you read the rebuttal and see if it clarified the issues identified in your review?
>
> Thanks
> Your AC.

---

> > ### Comment · AnonReviewer4 · 2020-04-02
> > **Thanks for the rebuttal response**
> >
> > I really appreciate the further explanation of the confusion in my review.
> > The reason for the empirical choice of 0.1 threshold seems valid in the visualization yet it is not very convincing because it changes the synthetic image even among the 4 visualized images. The threshold between 0.05 to 0.1 could be a good range of sensitivity study according to the image linked. If as the authors put "In our experiments, we found that the threshold was primarily useful for removing low-intensity background noise",  there is no comparison of the quality between images with and without thresholding or simply a good motivation to filter noises out if they are small. The value of 0.1 works for this dataset but not necessary for other datasets. A more general way to pick the threshold is rather needed.
> >
> > I believe this work is very promising but still, I am not convinced by the experiment choice. For example, the rationale for using % and decay. Why not directly compare a fully synthetic dataset to a real dataset? The explanation at 50%w/decay is observational rather explanatory.  Adding the real data in the training does not solve the imbalance issue since the real data is always used in training a classification task. Base on these, I decided to remain my original review as it was.

---

### Official Review · AnonReviewer2 · 2020-03-13
**Review: Synthesizing lesions using contextual GANs improves breast cancer classification on mammograms**

**Rating:** 3
**Confidence:** 4
**Recommendation:** Poster

**Summary:**

This paper presents a generative adversarial network for augmenting lesion or lesion-free mammogram images for addressing class imbalance issues in classification. Specifically, it proposes to use self-attentive and semi-supervised learning, considering contextual information in the breast tissue build on U-Net like architecture.

**Strengths:**

The problem is clearly stated. It sounds a good idea to synthesize more samples based on available mammogram data in clinical practice to solve the traditional class imbalance problem. Visual qualification for the results is well presented.

**Weaknesses:**

There are some unclear parts in the method section. It will be great this becomes clearer to readers: how the synthesized image could be made realistic. Following the self-attention module, it seems that the location of the generated lesion is based on the attention detected in the input image. Does this make sense all the time? Also, how to remove the existing lesion is not explained in the method.
In the experiment, I think it is meaningful to compare the classification accuracy between the proposed synthesized dataset and “balanced” but smaller real datasets to see the effectiveness of the proposed method.


**Justification Of Rating:**

Although this paper has a contribution to the research field, it still has room to improve and clarify (as detailed in Weaknesses). Especially, the ad hoc parts on page 4 could be revised for clarity.

**Paper Type:**

methodological development

**Questions To Address In The Rebuttal:**

I would like to see authors' responses related to the weaknesses above.

**Special Issue:**

no

---

> ### Author Response · Authors · 2020-03-28
> **Thank you for your review**
>
> Dear Reviewer,
>
> Thank you for taking the time to review our paper, we hope these responses sufficiently address your questions.
>
> 1. “There are some unclear parts in the method section. It will be great this becomes clearer to readers: how the synthesized image could be made realistic.”
>
>     We apologize for any lack of clarity. This is a great point: naturally, the adversarial nature of the GAN produces synthetic images that are hard to discriminate from real images. In addition, using self-attention modules and the U-net architecture helps to incorporate contextual information that produces lesions that match the background image features. However, given our target of improving classification accuracy, the best images for augmentation may not be the most “realistic”, ie. indistinguishable to the human eye from real images. Rather, they are the most useful for improving the classifier’s ability to classify cancer/no-cancer examples.
>
> 2. “Following the self-attention module, it seems that the location of the generated lesion is based on the attention detected in the input image. Does this make sense all the time?”
>
>     Correct, and more specifically, the self-attention module aids the network in selecting a candidate location that is more natural given the context. Whether the network successfully takes advantage of global context all the time is less clear; perhaps more specifically, the self-attention module is a mechanism that allows the network to be able to do so.
>
> 3. “Also, how to remove the existing lesion is not explained in the method. “
>
>     We apologize for this lack of clarity and will add a detailed description in the revision. In the case of removal, the generator outputs a pixel mask, which works as “white-out” over the lesion. When added to the base image, the lesion is covered up by this pixel mask. The lesion pixel values range from [-1.0, 1.0], so setting a threshold of <-0.1 allows the mask to subtract from the overall intensity of the lesion.
>
> 4. “I think it is meaningful to compare the classification accuracy between the proposed synthesized dataset and “balanced” but smaller real datasets to see the effectiveness of the proposed method.”
>
>     We note that we implement class balancing during training by randomly sampling positive and negative examples with equal probability at each training iteration. Creating a truly “balanced”, smaller dataset would amount to removing negative images to match the number of positives, but this would strictly mean decreasing the amount of data and is unlikely to improve overfitting on the positive cases. In other words, we would not expect such a dataset to outperform the baseline performance we report already using class balancing.

---

### Official Review · AnonReviewer3 · 2020-03-14
**Synthesizing lesions with GAN on mammogram patches for breast cancer classification**

**Rating:** 3
**Confidence:** 5
**Recommendation:** Poster

**Summary:**

The authors proposed a GAN-based method to synthesize and remove lesions on mammograms. The GAN model uses U-Net design with self-attention module and semi-supervised training loss. The authors augmented their original training set from OMI-DB dataset with the GAN-generated samples, and demonstrated improvement compared to their baseline model on patch-level malignancy classification on a test set of real mammogram data.

**Strengths:**

The paper is well written. Most methodological details are clearly documented, and well justified. Besides classification tasks, the authors also performed a t-SNE embedding analysis to understand how the real and synthetic data are clustered and the effect of the augmented model.

**Weaknesses:**

I am mostly concerned with the lack of comparisons in evaluation. There are a few papers on GAN-based synthesization of mammograms (for example, Ref [13]). The authors did compare their proposed method with any previous work. And there is no comparison to state-of-the-art breast cancer classification performance on OMI-DB dataset (besides a simple baseline model in the paper).

**Justification Of Rating:**

The paper is well written. Most methodological details are well justified. But it lacks comparisons to previous work (i.e. previous GAN-based methods for mammogram synthesization, and state-of-the-art breast cancer classification methods) in the evaluation.


**Paper Type:**

methodological development

**Questions To Address In The Rebuttal:**

1. The authors proposed to generate or remove lesions on top of the original image, instead of creating new samples from scratch. I wonder how does that compare to image infilling methods (for example used in Ref[13])?
2. It is unclear why validation set is evenly balanced with normal and malignant classes. AUC and other metrics (sensitivity, specificity, confusion matrix, etc.) on original data distribution would better reflect model performance on real world data.


**Special Issue:**

no

---

> ### Author Response · Authors · 2020-03-28
> **Thank you for your review**
>
> Dear Reviewer,
> Thank you for your feedback, we hope that we adequately addressed the points that you brought up.
>
> 1. “The authors proposed to generate or remove lesions on top of the original image, instead of creating new samples from scratch. I wonder how does that compare to image infilling methods (for example used in Ref[13])?”
>
>     We agree that this is a useful point to discuss. Infilling methods such as [13] require pixel-wise labeled data for training. Such data is more expensive to collect than bounding box labels and is also more rare. The work of [13] used the DDSM dataset, which does have pixel-wise segmentation masks, but DDSM is an old dataset of scanned film mammograms. The Optimam dataset used here contains modern digital mammograms and bounding boxes instead of segmentation masks. The Optimam dataset is also much larger than DDSM and is becoming the standard dataset for computer-aided detection in mammography.  Our method thus only requires bounding boxes over lesions without requiring pixel-level segmentation. Additionally, the U-net architecture allows the network to incorporate contextual information to generate lesions that match the feature statistics in the image. [13] relies on a randomly sampled lesion shape, which does not correlate with the feature statistics in the image.
>
> 2. “It is unclear why validation set is evenly balanced with normal and malignant classes. AUC and other metrics (sensitivity, specificity, confusion matrix, etc.) on original data distribution would better reflect model performance on real world data.”
>
>     The validation set is created using all of the data in our validation split (using all of the positive and negative data), and then simply oversampling the positive cases with augmentation. Because AUC is invariant to the positive proportion, we expect that the AUC does reflect model performance on real world data.

---

### Meta-Review · Area_Chair1 · 2020-04-05
**MetaReview of Paper25 by AreaChair1**

**Rating:** 2

**Metareview:**

The paper has been reviewed by three experts in the field that list a fair number of positive and negative points.  On the positive side, they all agree that the paper addresses an important problem and is well written.  On the negative side, the paper seems to have very little technical/methodological contribution beyond the combination of well-understood techniques applied to the problem of mammogram generation.  Such issue could be compensated by a careful ablation study and thorough comparisons with other similar data augmentation methods recently proposed, but these two points are addressed only partially by the paper.  Initial reviews were quite balanced between accept and reject, but after rebuttal, the reviewers seem to be leaning towards rejection.  My opinion is that the paper is good, but it needs to address the issues identified by the reviewers  before it can be published.

**Paper Type:**

both

**Special Issue:**

no

---

### Decision · Program_Chairs · 2020-04-11

Reject